# Position: We Need An Algorithmic Understanding of Generative AI

Oliver Eberle [1 2]   Thomas McGee [3]   Hamza Giaffar [4]   Taylor Webb [5]   Ida Momennejad [5]

## Abstract

What algorithms do LLMs actually learn and use to solve problems? Studies addressing this question are sparse, as research priorities are focused on improving performance through scale, leaving a theoretical and empirical gap in understanding emergent algorithms. This position paper proposes *AlgEval*: a framework for systematic research into the algorithms that LLMs learn and use. AlgEval aims to uncover algorithmic primitives, reflected in latent representations, attention, and inference-time compute, and their algorithmic composition to solve task-specific problems. We highlight potential methodological paths and a case study toward this goal, focusing on emergent search algorithms. Our case study illustrates both the formation of top-down hypotheses about candidate algorithms, and bottom-up tests of these hypotheses via circuit-level analysis of attention patterns and hidden states. The rigorous, systematic evaluation of how LLMs actually solve tasks provides an alternative to resource-intensive scaling, reorienting the field toward a principled understanding of underlying computations. Such *algorithmic explanations* offer a pathway to human-understandable interpretability, enabling comprehension of the model's internal reasoning performance measures. This can in turn lead to more sample-efficient methods for training and improving performance, as well as novel architectures for end-to-end and multi-agent systems.

[1]Technische Universität Berlin, Berlin, Germany [2]BIFOLD–Berlin Institute for the Foundations of Learning and Data, Berlin, Germany [3]University of California Los Angeles, Los Angeles, USA [4]Halıcıoğlu Data Science Institute, University of California San Diego, San Diego, USA [5]Microsoft Research NYC, New York, USA. Correspondence to: Ida Momennejad <idamo@microsoft.com>, Oliver Eberle <oliver.eberle@tu-berlin.de>.

*Proceedings of the 42^{nd} International Conference on Machine Learning*, Vancouver, Canada. PMLR 267, 2025. Copyright 2025 by the author(s).

## 1. Introduction

Large language models (LLMs) have soared to prominence, yet a fundamental question remains: What algorithms do LLMs actually use to solve problems? As the "gold rush" of scaling prioritizes practical breakthroughs, research priorities have centered on improving performance through scale, often regardless of guarantees or costs, while interpretability efforts have largely focused on understanding isolated mechanisms. Algorithmic understanding is often left behind. **This position paper argues that the ML community should prioritize research on an *algorithmic* understanding of generative AI**.

Existing work on understanding algorithmic operations in LLMs (Zhou et al., 2024; Li et al., 2023; von Oswald et al., 2024; Yang et al., 2024), while impressive, remain surprisingly few in number. Recent interpretability research has prioritized the exploratory analysis of low-level circuit mechanisms (Olah et al., 2020; Olsson et al., 2022), often without clear hypotheses, and even position papers that note the importance of algorithmic understanding only mention it as one among many other directions (Vilas et al., 2024). Algorithmic research on LLMs has taken a backseat among priorities, with theoretical work on the topic almost entirely lacking for both individual and multi-agent LLM systems. While multi-agent systems are now common solutions for reasoning and planning with Transformers (Wu et al., 2023a; Webb et al., 2024; Nisioti et al., 2024), theoretical foundations for efficiently building them remain underexplored.

While scaling has led to impressive results on a wide range of tasks, its limits remain unclear. Scale, rather than hypothesis-driven methods, has become the prevailing drive of general-purpose architectures since the rise of deep learning in the 2010s. This *Bitter Lesson* (Sutton, 2019), combined with the hypothesis that reward may be enough for the emergence of intelligence (Silver et al., 2021), have led to an emphasis on data- and compute-heavy approaches, prioritizing training data and fine-tuning with existing architectures. This trend has also widened the gap between frontier models and interpretability research, severely limiting their transparency, trustworthiness, and compliance with AI regulations (Samek et al., 2021; Kaur et al., 2022).

The field is increasingly encountering the limitations of available data and its quality, and the rising computational

costs and diminishing returns of scale (Villalobos et al., 2024). This is in contrast to biological intelligence and brains, which provide an existence proof for a far more data- and energy-efficient approach. Recent studies suggest that optimizing inference-time compute can be more beneficial than simply scaling parameters, prompting shifts from feed-forward parameter growth to inference-time compute (Snell et al., 2024). Thus, given the environmental costs, scaling without understanding is not a sustainable path forward, particularly for multi-agent AI, where insights into system interactions are increasingly crucial.

This position paper calls for prioritizing systematic research on the algorithmic understanding of generative AI. An algorithm is typically defined as a finite set of rules or operations for transforming inputs into outputs (Turing, 1936; Knuth, 1968), which can be combined to form efficient strategies to solve larger problems. Algorithmic explanations of LLMs, therefore, involves uncovering the specific step-by-step procedures or "computational primitives" these models effectively learn and execute during task solving. By examining operations within a model's architecture, its parameters, and inference process, we can comprehensively determine how the model arrives at its outputs.

A systematic framework for algorithmic understanding should address: a) What algorithms can generative AI learn, and how does this depend on factors such as model size, training data, fine-tuning, and in-context learning? b) Are there provable guarantees for any such algorithmic abilities? c) How can we build multi-agent systems in order to implement specific algorithms? d) How can we set algorithmic objectives for training and fine-tuning? e) How can we create a repository of algorithmic abilities? f) How can we study the selection and composition of these components to solve prompted tasks? g) How can we design architectures to guarantee specific algorithmic capacities? In what follows, we outline *AlgEval*, a research program for algorithmic evaluation and understanding of generative AI.

## 2. Related Work

To capture the rich internal computations implemented by increasingly complex machine learning (ML) systems, efforts in explainable AI and mechanistic interpretability have shifted focus to the inner workings of generative models, introducing approaches to uncover internal circuits (Olah et al., 2020; Wang et al., 2023), representations (Todd et al., 2024), dynamical motifs (Yang et al., 2024), and computational subgraphs (Schnake et al., 2022; Geiger et al., 2022), laying the foundation for an *algorithmic understanding* of model predictions.

Interpretability research initially emerged for deep classification models, before the era of generative AI (Lipton, 2017;

Montavon et al., 2018). Understanding classification models has largely focused on identifying relevant input features or heatmaps at intermediate layers, using methods such as perturbation-based (Lundberg & Lee, 2017), attention-based (Abnar & Zuidema, 2020), and gradient-based (Baehrens et al., 2010; Sundararajan et al., 2017; Ali et al., 2022) approaches. With the shift toward generative AI and sequence modeling, a principled understanding of internal processes, beyond input-output relations or isolated mechanisms, has become essential. This presents a significant technical challenge due to the scale and complexity of today's frontier models. Thus, some researchers have focused on smaller or synthetic language models for targeted analysis and empirical studies, e.g., GPT-2 small (Wang et al., 2023; Conmy et al., 2023; Hanna et al., 2024) or toy Transformers (Liu et al., 2023; Ye et al., 2025).

On the other hand, early efforts to analyze LLM mechanisms focused on localizing specific functions within isolated model components (Vig & Belinkov, 2019; Clark et al., 2019), such as individual neurons (Gurnee & Tegmark, 2024; Zhou et al., 2018; Templeton et al., 2024), or individual attention heads (McDougall et al., 2024). More recent work has investigated how these components combine to form functional circuits (Olsson et al., 2022; Wang et al., 2023; Tigges et al., 2024), while other work has characterized the representations that support some higher-level computations, e.g., function vectors (Todd et al., 2024). Probing techniques have further been developed to assess whether specific properties can be accurately decoded from a model's latent representations (Conneau et al., 2018; Hewitt & Manning, 2019), particularly in the analysis of reasoning strategies (Ye et al., 2025). While these approaches move toward a more integrated understanding, current findings are still largely fragmented, and we lack a solid theoretical foundation for understanding how these various components come together to implement algorithms.

## 3. AlgEval: Toward Algorithmic Evaluation and Understanding of LLMs

Algorithms consist of modular subroutines that exhibit compositionality, allowing them to be reused and recombined into efficient strategies for solving increasingly complex problems. From this conceptual starting point, AlgEval proposes a path toward algorithmic evaluation and understanding of LLMs through algorithmic primitives and their composition, analogous to a vocabulary and grammar. We then explore methods to evaluate them, from the common analysis of attention weights, latent representations, and circuit methods, to new approaches like inference-time compute and evaluating alternative solutions.

A key challenge in the algorithmic understanding of LLMs is designing tasks that are complex enough to support spe-

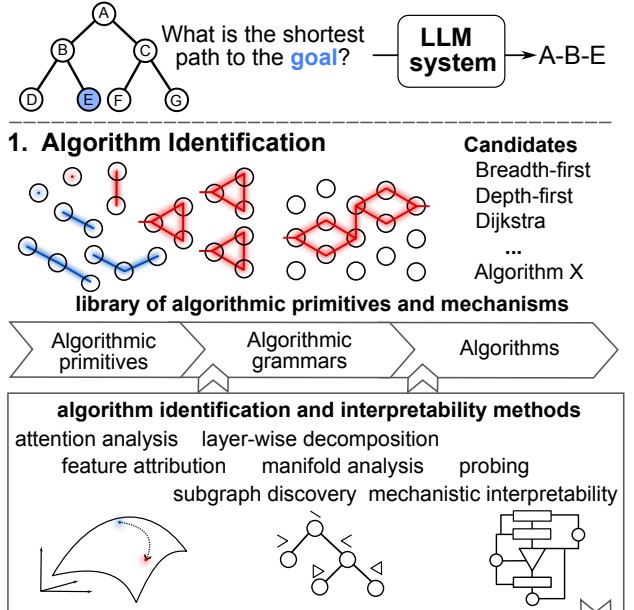

## 1. Algorithm Identification

**Candidates**
Breadth-first
Depth-first
Dijkstra
...
Algorithm X

**library of algorithmic primitives and mechanisms**

Algorithmic primitives → Algorithmic grammars → Algorithms

**algorithm identification and interpretability methods**

attention analysis   layer-wise decomposition
feature attribution   manifold analysis   probing
subgraph discovery   mechanistic interpretability

## 2. Empirical Verification and Theoretical Analysis

| performance | algorithmic desiderata | theory |
|---|---|---|
| task accuracy | faithfulness | learning theory |
| generalization | algorithmic complexity | algorithmic theory |
| error analysis | computational efficiency | complexity |

## 3. Improved Design and Insights

| model development | data benchmarks | theoretical frameworks | algorithm discovery |
|---|---|---|---|
| improving and building gen. AI | progress measure reproducibility | proofs guarantees | novel algorithms scientific insight |

*Figure 1.* AlgEval: A methodological path to prioritizing algorithmic evaluation and understanding of LLMs.

cific algorithmic hypotheses and for which ground truth responses and strategies are available. An example is goal-directed navigation on deterministic graphs, Figure 1 (top). To solve such a task, classical search algorithms like breadth-first search (BFS), Dijkstra, or depth-first search (DFS) offer a verifiable algorithmic ground truth for evaluation. Moreover, heuristic, optimization-based approaches like simulated annealing (Kirkpatrick et al., 1983; Cerný, 1985; Metropolis et al., 1953), or amortized inference (Gershman, 2019) could be at play. For instance, in amortized inference or "learning to infer," rather than solving problems from scratch, the model compresses frequently used inference routines into a parametric function. In AlgEval, these hypotheses guide model analysis by evaluating internal mechanisms and inference-time generation to extract functional structure.

This hypothesis-driven approach re-centers the understanding and development of LLMs on scientific principles, emphasizing rigor in evaluating learned mechanisms. Drawing from Marr's three levels of analysis, i.e., computational, algorithmic, and implementation (Marr, 1982; Vilas et al.,

2024), AlgEval prioritizes understanding mechanisms at the algorithmic level over their physical realization. By systematically testing algorithmic hypotheses, we also move beyond the assessment of behavioral goals at the computational level to uncover internal structures that enable robust problem-solving. Next, we describe what we refer to as *algorithmic primitives*, and how models can piece them together to form algorithmic solutions.

### 3.1. Algorithmic Primitives and Vocabulary

**Basic primitives.** We define primitives to be the basic elements necessary to realize a specific algorithm. By iteratively breaking down an algorithm, a vocabulary of essential primitives can be obtained. In the context of LLMs, a variety of such primitives have been identified through methods from interpretability and data analysis. These approaches have led to the discovery of vector-based mechanisms, including function vectors for in-context learning mappings (Todd et al., 2024; Edelman et al., 2024), vector arithmetic computations (Merullo et al., 2024), steering vectors (Yang et al., 2024), copy suppression (McDougall et al., 2024), and key-value memory retrieval (Geva et al., 2021). Additional primitives have been described that identify duplicate elements, and inhibit or increase attention to specific sequence elements (Wang et al., 2023).

The functions of algorithmic primitives, as we defined them, should be predominantly domain-general and applicable to a wide range of sequence modification tasks. While the interpretability community provides a starting point to discover possibly domain general primitives, e.g., in-context learning (ICL) (Elhage et al., 2021; Olsson et al., 2022), token binding (Vasileiou & Eberle, 2024; Feng & Steinhardt, 2024), or recognizing semantic relationships (Ren et al., 2024), principled frameworks for connecting and combining these individual findings are mostly lacking. On the other hand, inspired by the Transformer architecture, a framework of programming primitives, termed 'restricted access sequence processing language' (RASP) (Weiss et al., 2021; Zhou et al., 2024), has been developed to formalize the study of algorithmic implementations in Transformers' interpretable sequence operations, including basic select and aggregate manipulations. While helpful, RASP focuses on what could be built with LLMs in minimal scenarios, currently remaining inapplicable to understanding real world generative AI.

A key goal of AlgEval is the identification and evaluation of algorithmic primitives as clearly defined operations that can be assessed and validated at a lower level of complexity, and bridged to more sophisticated algorithmic levels. Similar to current discussions on the emergence of universal representations (Huh et al., 2024), this enables building a systematic catalog of universal primitives across models and tasks.

**Primitives as foundational algorithms.** Essential primitives can be composed into domain-general basic algorithms for sequence generation tasks, leading to the discovery of algorithmic subgraphs and circuits that integrate multiple primitives and mechanisms. A number of disjointed findings are noteworthy. For instance, a network of induction, inhibition/excitation, duplicate token, and copy heads have been identified to solve the task of indirect object identification (Wang et al., 2023). Different algorithms have been discovered that solve modular arithmetic via combinations of circular embeddings and simple trigonometric operations (Nanda et al., 2023; Zhong et al., 2023), which can be understood through different analytical solution approaches. By combining memory heads, that promote internally stored information, with in-context heads, a mechanism for factual recall has been reconstructed (Yu et al., 2023). We consider these efforts as starting points for studying simple combinations of primitives. On the other hand, a set of theoretical studies focusing on complexity analysis (Elmoznino et al., 2024), algorithmic selection and assembly theory (Sharma et al., 2023), and combining and analyzing primitives via structured interactions (Morris et al., 2019; Eberle et al., 2022; Schnake et al., 2022; Fumagalli et al., 2023), call for an integration of theoretical and empirical contributions toward algorithmic discovery.

**Algorithms as primitives.** Recent studies have demonstrated how Transformers can display classic ML methods, including kernel-based approaches (Tsai et al., 2019), support vector machines (Tarzanagh et al., 2023), Markov chains (Zekri et al., 2024), higher-order optimization methods for ICL (Fu et al., 2024), and temporal difference learning (Demircan et al., 2024). Further research is required to understand whether they can be understood as algorithmic primitives that can be combined, further broken down into more basic primitives, or both. This highlights the need to understand the extent to which primitives serve as task-specific or domain-general building blocks, and whether we can identify a systematic hierarchy of primitives.

### 3.2. Algorithmic Composition

**Compositionality.** The combination of algorithmic primitives into more complex algorithms is a form of *compositionality* that, in principle, should support the construction of a vast number of combinations from a finite vocabulary. The search for algorithms is thus related to the broader debate about the extent to which LLMs are capable of compositional reasoning and generalization. While there is some evidence to support the existence of compositional representations and mechanisms in generative models (Lepori et al., 2023; Campbell et al., 2024), there is also evidence for persistent failures in tasks that require compositional reasoning (Lewis et al., 2022; Mitchell et al., 2023; Conwell et al., 2024). Studying the algorithms used by these models

can clarify whether their reasoning is truly compositional or reflects ineffective strategies like memorization (Power et al., 2022; Qiu et al., 2022), or instabilities in representing rare events (Kandpal et al., 2023).

An algorithm's relevant primitives can be combined through several strategies. Previous works have explored optimization-based methods, such as learned graph structures and message passing (Geiger et al., 2022), that enable dynamic interaction among components, while symbolic and compositional approaches provide explicit modular frameworks for task-specific solutions (Geiger et al., 2022; Wu et al., 2023b). ICL techniques, like providing grammar through input (Galke et al., 2024), can steer model behavior based on contextual cues. In the context of RASP, hard-coded aggregate functions can be used to nest several primitives, forming more complex functions, e.g., able to reverse or sort sequences (Weiss et al., 2021). So far, it remains unclear whether compositionality emerges when learning next-token prediction on, for example, a trillion tokens of compositional data, and to what extent it can be built into the architecture or training methods.

To foster deeper understanding of algorithmic composition, we need targeted empirical studies combined with building compositionality from first principles. Promising starting points include imposing known compositional constraints on the grammar, e.g., via hierarchical pyramids (Lin et al., 2017), equivariant structure (Satorras et al., 2021), conditions imposed by the data-generating function (Wiedemer et al., 2023), or algorithmic complexity (Elmoznino et al., 2024). Assembly theory offers an evolutionary perspective on forming complex algorithmic grammars by selecting and recombining primitives (Sharma et al., 2023).

### 3.3. Methodologies for Algorithmic Evaluation

AlgEval focuses on methodological advances necessary to identify, evaluate, and discover algorithms. This motivates combining existing interpretability techniques with novel approaches to capture the complexity of modern AI.

**Analyzing representations and attention.** Neural representations, patterns of activity across neural populations or intermediate Transformer layers, are increasingly recognized as key to understanding or modifying network computations (Zou et al., 2023; Sucholutsky et al., 2024). Analyses of representational similarities within or across layers, using an array of similarity measures sometimes inspired by cognitive science and neuroscience, elucidate how these layers transform information (Kornblith et al., 2019; Klabunde et al., 2024; Yousefi et al., 2024; Sucholutsky et al., 2024; Williams et al., 2021; Giaffar et al., 2024). For instance, LLM embeddings can reveal interpretable structures for deception detection by identifying a 2D subspace encoding true/false statements (Bürger et al., 2024), and a three-stage

process of deceptive behavior has been uncovered through low-dimensional projections (Yang et al., 2024).

Complementary to representation analyses, attention in LLMs is commonly analyzed to identify message passing operations among tokens. Layer-wise attention scores help interpret token importance and internal model structures, including attention rollout and attention flow (Abnar & Zuidema, 2020). To capture task-specific model processing, feature attribution methods compute feature importance scores, addressing the limits of attention analysis in providing faithful explanations (Wiegreffe & Pinter, 2019), with techniques like saliency (Chefer et al., 2021) and modified gradient methods (Ali et al., 2022; Achtibat et al., 2024; Jafari et al., 2024). Furthermore, internal analysis of attributions can aid in discovering relevant representational concepts (Kauffmann et al., 2022; Chormai et al., 2024).

The integration of attention and representation analyses is a key feature of AlgEval, as information passing between tokens and representation analyses can uncover network structures and transformations, helping us understand algorithmic primitives and compositionality in problem-solving. (see Section 4).

**Subgraphs and circuits.** The identification of relevant internal structure presents a current methodological frontier in our understanding of LLMs. Various methods have been proposed to extract circuits (Olah et al., 2020; Wang et al., 2023), subgraphs (Schnake et al., 2022; Geiger et al., 2022), feature interactions (Eberle et al., 2022; Fumagalli et al., 2023; Vasileiou & Eberle, 2024; Kauffmann et al., 2024), and causal symbolic models (Geiger et al., 2022). Key techniques for discovery of internal structure include activation patching (Wang et al., 2023), automatic circuit discovery (Conmy et al., 2023), attribution patching (Syed et al., 2024; Hanna et al., 2024), and graph explanations (Schnake et al., 2022; Sanford et al., 2024). Viewing LLMs as graphs provides a complementary perspective of sequence processing as computations that extend across multi-hop neighborhoods that form substructures (Besta et al., 2024), build motifs and compute higher-order interactions across neurons (Eberle et al., 2022; Schnake et al., 2022; Fumagalli et al., 2023).

### 3.4. Identifying and Structuring Primitives

Methodologically, we have identified the following five steps as crucial components of AlgEval: (a) **Identify and form a library of primitives** which can grow over time and anchor corresponding tasks and mechanisms. Each algorithmic primitive can correspond to multiple tasks and algorithms, supporting different mechanistic implementations. Primitives can be either hypothesis-based, rooted in decades of theoretical algorithm research, or empirically observed. (b) **Build a collection of simple tasks** which require a set of primitives for their solution. Examples include sequence induction (Olsson et al., 2022), or copying a sequence of unique tokens (Zhou et al., 2024). (c) **Create a library of mechanisms** that implement primitives, along with corresponding interpretability and analysis tools to identify them as discussed in Section 3.3. (d) **Analysis of Composition** is crucial for identifying primitives, as it involves understanding how they combine, how algorithms are implemented across layers and inference, and whether compositional patterns generalize across tasks and models. (e) **Ablations** serve to identify and evaluate primitives' role, which necessitates developing tools for causal intervention and ablation (Geiger et al., 2022; Talon et al., 2024), as well as using statistical tests. Newly discovered primitives, along with their associated tasks, mechanisms, and methods, are then added to the growing library of primitives for future analyses and integration into new models.

### 3.5. New Directions for Algorithmic Analysis

**The role of in-context learning.** Recent work has explored how ICL improves transformer performance (Fu et al., 2024), one proposing Transformers are algorithms (Li et al., 2023). A recent paper analyzed changes in attention and representation due to ICL and how it related to improvements in behavior (Yousefi et al., 2024). While that work did not focus on interpreting the effect of these changes on algorithmic primitives or composition, one important future direction will be to analyze the algorithmic consequences of ICL, such as promoting the use of specific algorithms.

**Inference-time compute.** Inference-time compute has recently arisen as a new paradigm for reasoning with LLMs. In this approach, rather than solving a problem via a single feedforward pass, the autoregressive outputs of the model can be used to perform intermediate computations. Examples of this approach include chain-of-thought (Wei et al., 2022), explicit tree search (Yao et al., 2024), agent-based approaches (Webb et al., 2024), and models such as o1 (Jaech et al., 2024) and the open-source R1 (DeepSeek-AI et al., 2025) that are trained to perform inference-time compute via amortized optimization. Some have even argued for scaling inference-time compute instead of scaling parameters or training (Snell et al., 2024).

AlgEval also applies to emergent algorithms at inference time, where sequential outputs can be more amenable to algorithmic analysis than high-dimensional feedforward computations. For example, LLMs can be trained to explicitly search via their outputs, implementing search procedures like exploration and backtracking through traces provided in context (see *in-context search, SearchFormer, Stream of Search*) (Gandhi et al., 2024; Lehnert et al., 2024). This is particularly revealing when models acquire algorithms through amortized inference for downstream tasks rather than via supervised fine-tuning or ICL, potentially yielding

novel or emergent solutions.

While no existing work systematically examines inference-time emergent algorithms, likely due to their novelty, the core AlgEval principles of identifying algorithmic primitives, forming top-down hypotheses, and bottom-up testing remain applicable. There may also be interactions between feedforward algorithms and inference-time compute, where certain processes like search are offloaded to the output space, allowing the feedforward pass to specialize in different primitives. Finally, it is crucial to ensure causal linkage between inference-time outputs and actual performance, since chain-of-thought can be unfaithful or unrelated to the model's real reasoning (Turpin et al., 2024; Stechly et al., 2024).

**Reinforcement learning and memory compression.** A recent open-source LLM, DeepSeek R1 (DeepSeek-AI et al., 2025), used reinforcement learning (RL) and inference-time compute to match the performance of OpenAI's o1 at a fraction of the training cost. Interestingly, this model displayed an apparently emergent form of backtracking (referred to as an 'aha moment'). A future direction of research is to study to what extent this behavior emerged purely as a consequence of training with RL, as opposed to relying on documents in the base model's training data that include similar examples of backtracking (annotated math solutions). Given o1 was supposedly also trained on annotated math solutions, it is important to study how training with RL, particularly Group Relative Policy Optimization (GRPO) (Shao et al., 2024), is necessary to elicit this behavior. Open-source models like DeepSeek R1 offer an opportunity to study how training data and RL shape their algorithmic vocabulary and compositions thereof. A key question herein is whether RL led to cached strategies for amortized inference, enabling "learning to infer," by compressing prior inferences (Gershman, 2019; Radev et al., 2020). Such strategies can link learning to compressed memory representations, which is also observed in the human brain and behavior (Momennejad et al., 2017; Russek et al., 2017; Momennejad, 2020; Brunec & Momennejad, 2021; Momennejad, forthcoming).

## 4. Case Study

To ground our position in an empirical example, we conducted a case study focused on LLMs, which have been shown to perform poorly on graph navigation and multi-step planning tasks (Momennejad et al., 2023). In cases where they do succeed, it remains unclear how they solve these problems, e.g., whether they implement classic search algorithms or use other strategies. To address this question, we studied the algorithms used by widely used LLMs, instruction-tuned Llama-3.1 with 8B and 70B parameters, in the context of graph navigation. We considered a simple tree graph structure, presented in a prompt that describes

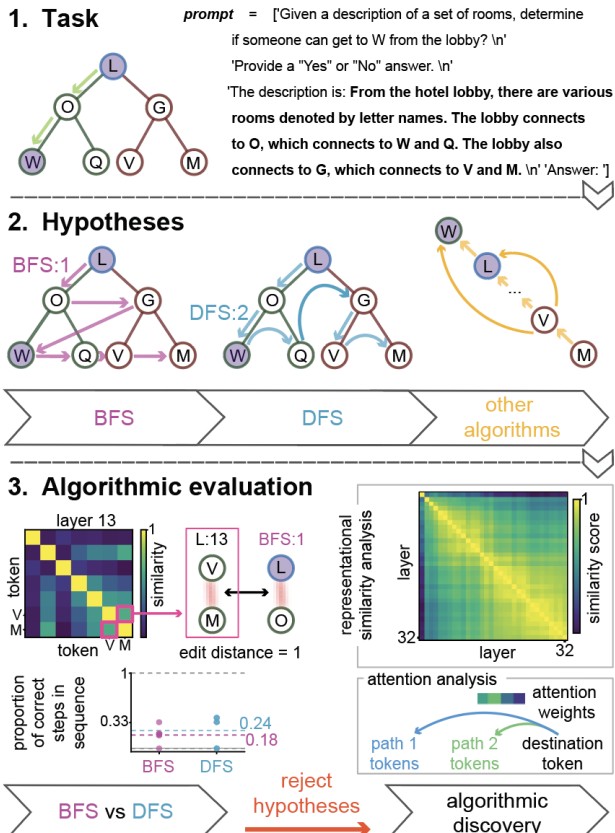

*Figure 2.* Case study. A graph navigation task, algorithmic hypotheses (possible rollouts for DFS and BFS are shown), and potential methods for algorithmic evaluation.

how rooms (nodes) connect to one another (edges) and tasks the model with determining whether a direct path from the start to the goal node exists (Figure 2).

A straightforward hypothesis to test is that the model may use standard search algorithms like DFS, BFS, or Dijkstra to find paths between graph nodes, with each layer potentially representing one search step. This analysis assumes that each layer corresponds to one visited node or node pair and that multiple connections may be evaluated simultaneously. If the layers successfully identify the correct path, we can infer its sequence of visited nodes by examining which node tokens receive the highest attention or exhibit the strongest representational similarity across layers (Kriegeskorte et al., 2008; Kornblith et al., 2019; Manvi et al., 2024). These combined analyses of attention and representation provide insight into the potential step-by-step algorithmic procedures underlying the LLM's graph exploration.

**Prompt.** We introduce the model to a two-step tree graph following the prompt from Momennejad et al. (2023), which demonstrated that LLMs struggle with graph navigation and especially tree search. The model is tasked with determining the validity of a given path, producing a single token output: 'yes' or 'no'. The full prompt and task for starting from

the 'lobby' and goal location W are shown in Figure 3a. Note that some nodes appear multiple times in the prompt (e.g., 'lobby' is repeated four times), thus in our analysis in Figure 3c, we display the representation trajectory for each appearance of a node.

## 4.1. Cascading Attention Analysis

Analyzing layer-wise attention matrices in LLMs (Vaswani, 2017) provides a direct and commonly used way to trace message-passing operations among tokens. To better understand how the model's algorithm leverages attention, we analyzed 1) the attention from each graph node to its preceding nodes and the goal, and 2) the attention from the final token to all nodes. The former allowed us to test how the graph nodes "search over themselves" across layers, while the latter allowed us to test how the model searches over relevant tokens at inference time. We next present results on Llama-3.1-8B with additional analyses of the 70B model presented in Appendix A.4.

To test whether the model performs search over graph nodes across layers, we analyzed how the final token's attention shifts between correct and incorrect paths as shown in Figure 3. We first used a linear mixed-effects model with logit-transformed average attention as the outcome variable, pathway (correct or incorrect) as a fixed effect, and layer number as a random intercept. Results indicated that the final token allocated significantly more attention to rooms in the correct pathway than to those in the incorrect one ($b = 0.33$, $SE = 0.07$, $t(2015) = 4.51$, $p < .001$) (Appendix Table 1). The model included a random intercept for layer number (variance $= 0.96$, $SD = 0.98$), and the residual variance was 2.80 ($SD = 1.67$). A layer-wise analysis showed that the final token allocated significantly more attention to rooms on the correct path in 14 of 32 layers (Appendix Table 2), with only three layers exhibiting significantly more attention to the incorrect path (Appendix Table 3).

Analyzing attention to individual nodes, response tokens, and the goal token revealed an interpretable layer-by-layer sequence leading to the correct response: early-to-mid layers attended to pairwise node links, while attention to the goal node W peaked in layers 13–14, and the final token's attention shifted to the correct response around layer 19.

The cascading attentional spread from each node to its predecessors may incrementally direct attention to the correct path, suggesting a **policy-dependent** algorithm rather than an exhaustive search like BFS or DFS. The model seems to (1) incrementally attend to the path leading to the goal via query-key attention weights, then (2) attend to the goal token in later layers as presented in Figure 3.

## 4.2. Analysis of Feedforward Representations

To characterize how graph representations contained in feedforward activity change across LLM layers, we first defined the token-by-token representational similarity matrices $V^i$ for each layer, indexed $i$, entries of which $V_{xy}^i = u_{i,x}^T u_{i,y}$ are inner products between activation vectors $u_{i,x}$ and $u_{i,y}$, for room tokens $x$ and $y$ respectively. To ask if we could identify discrete changes in the graph representational geometry between neighboring layers that might be interpreted as steps of an algorithm, we computed layer-by-layer similarity matrices, $S^d$ using similarity measure $d$ (Kornblith et al., 2019; Williams et al., 2021), where $S_{ij}^d = d(V^i, V^j)$. From layers 4 to 32, the representational similarity between neighbouring layers remains high (with scores above 0.95), suggesting that graph representational geometry changes relatively smoothly from early to late layers and does not appear to change in a clear step-like fashion as shown in Appendix Figure 6.

**Comparing LLM vs. hypothesis sequences.** To compare the model's feedforward activations to classical search algorithms, we extract a possible sequence of algorithmic steps from the LLM hidden unit activations: for each layer $i$ we identified the node pair $e_i = (x, y)$ with highest representational similarity in $V_i$ (Figure. 2). We compared this sequence of edges with all possible unique rollouts of BFS and DFS by: i) computing the edit distance between each LLM step and each BFS/DFS trajectory, and ii) finding the longest sequence of correct steps for each trajectory; to be maximally permissive, we only required LLM steps to be in the correct sequence, not necessarily in adjacent layers. This analysis revealed that the sequence of steps identified in LLM layers is not well matched to any full trajectory under either BFS or DFS (the mean proportion of correct matching steps in sequence is 0.18 for BFS and 0.24 for DFS; Figure 2).

**Competing representations.** We next analyzed the evolution of node representations across layers. In Figure 3c, we present a low dimensional latent space (t-SNE) projection of node representations across all layers, along with the final end-of-sequence ('eos') representation ('yes' or 'no'). Colors represent room appearances (graph nodes) in the prompt, and numbers indicate the layer from which the representation is extracted, showing the evolution of representational distance among rooms over layers.

While in the first layers all nodes are closely clustered together, across the layers, we observe a clear progressive separation of tokens associated with the 'lobby' from other nodes (Figure 3c). This hints at an algorithmic strategy that distinguishes between the anchor tokens (the 'lobby') and a set of varying potential goal nodes. Interestingly, we find further structure developing across the latter graph nodes: a consistent clustering of non-goal nodes, and a grouping of

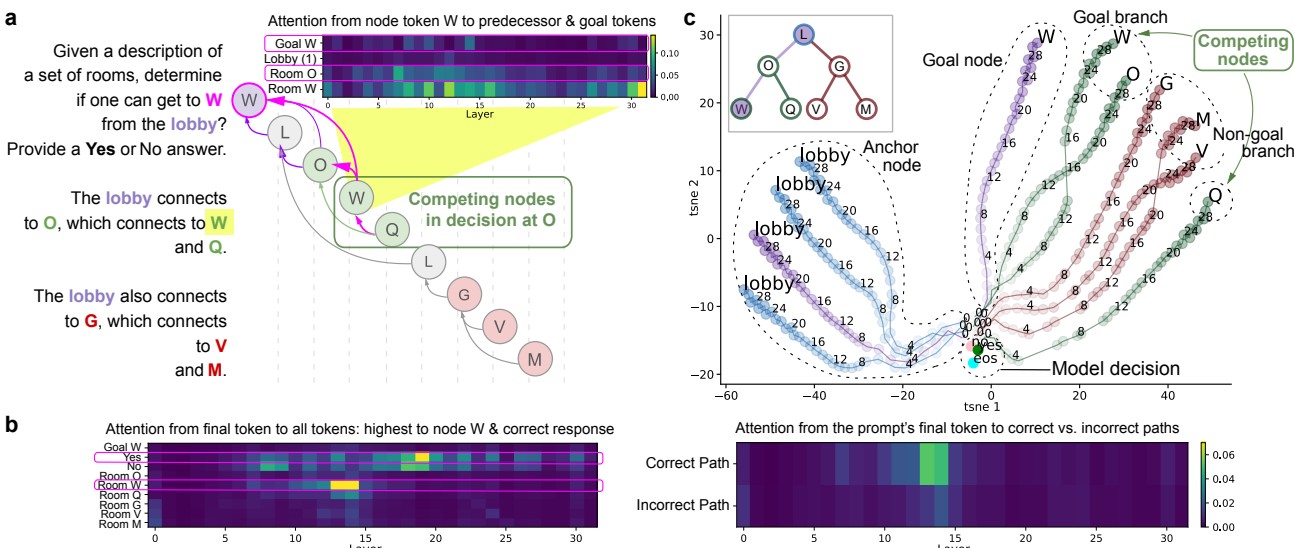

*Figure 3.* Case study on algorithmic discovery. (**a**) Attention heatmap: from goal to the correct vs. incorrect pathway. (**b**) Attention is mostly allocated to the goal location M, followed by V and G on the correct path. (**c**) Separation of room representations across layers.

subsets among them as shown in Appendix Figure 5. Similar to our attention analysis results (Figure 3a), we observe that the competition between the goal node W and Q, W's closest competitor node in the same branch, is reflected in their progressively increasing separation in representation space (see Figure 3c and Appendix Figure 5). This separation typically emerges at intermediate layers and follows the high attention that W receives from the final 'eos' token at these stages (Figure 3b). Lastly, we observe that the representation of the goal token, defined in the prompt context, consistently maps onto a distinct trajectory, separating its representation from all other nodes while being closest to the 'lobby' representation (purple goal branch in Figure 3c).

**Search strategies in larger models.** To test whether increased scale results in a different, potentially more robust algorithmic search strategy, we repeated our experiments on a ten times larger model (Llama-3.1-70B-Instruct). As shown in Figure 7 and Figure 9 in the Appendix, our results are consistent to those of the smaller model, displaying similar patterns in both representation space and attention scores, but with a more pronounced separation between correct and incorrect trajectories. We further compared the sequence of states with the highest representation activation across layers to potential search sequences generated by BFS and DFS strategies, and found no differences in search strategy between the models as shown in Figure 8 in the Appendix. Neither sequence of representations closely matched the BFS or DFS rollouts on the target graph.

**Interpretation.** Using the framework of AlgEval, we found that attentional and representational patterns in LLMs do not align neatly with classical search algorithms, highlighting the role of algorithmic explanations in making such

discrepancies interpretable. The attention patterns we observed (Figure 3b and Figure 9 in the Appendix) suggest that current models do not construct a full world model or perform exhaustive search. We further find that representations of graph nodes evolve layer by layer, increasing the representational distance among key nodes, e.g., the distance between the closest competing node and the goal node (W vs. Q, Figure 3 and Figure 7). Future studies should verify whether this competition-driven separation across layers, also observed in simpler settings (Wang et al., 2023), represents an algorithmic primitive, and whether it is driven by inhibition or mover attention heads, function vectors (Todd et al., 2024), or other mechanisms. Furthermore, it remains to be seen to what extent a model's search strategy varies with task and model complexity, especially in relation to its failure modes when navigating more complex graphs.

## 5. Alternative Views

Our proposal has some overlap with mechanistic interpretability (Olah et al., 2020; Olsson et al., 2022), but differs with key distinctions. First, mechanistic interpretability often emphasizes bottom-up perspectives, even advocating for hypothesis-free circuit analysis (Olah, 2023). Second, while it is good to remain open-minded about hypotheses, it has been argued that interpreting data with an 'innocent eye' is not possible (Gershman, 2021). Thus, we advocate for combining top-down algorithmic hypotheses with bottom-up evaluation. Third, while mechanistic interpretability focuses on low-level circuits, AlgEval targets the algorithmic level of explanation (Marr, 1982), integrating primitives and their composition, from circuits to higher-level computations. This perspective also connects to work on neural

algorithmic reasoning (Veličković & Blundell, 2021), which aims to integrate algorithmic structure into neural architectures by operating in high-dimensional latent spaces while performing computations aligned with a specific target algorithm. Fourth, we contrast our approach with "AI-assisted interpretability," in which AI systems explain other AI systems (Choi et al., 2024; Li et al., 2024a; Olah, 2023). Although it has produced intriguing results, like automated neuron descriptions (Choi et al., 2024), we maintain it cannot replace hypothesis-driven research. Designing rigorous algorithmic tests likely requires reasoning skills beyond current models, so human involvement remains vital. Finally, AlgEval may offer an alternative to the dominant scaling paradigm (Sutton, 2019) by systematically understanding emergent algorithms and embedding them into architectures, rather than relying solely on training data and scale.

## 6. Discussion and Future Directions

In this position paper, we advocate for systematic research into algorithms learned and used by generative AI. We introduced AlgEval as a framework to investigate algorithmic primitives, their composition, and the impact of architecture, parameters, and optimization. AlgEval extends to inference-time compute and how it depends on training data and objectives, with implications for both empirical design and theoretical understanding.

**Theoretical research.** Recent theoretical work has addressed hierarchical language learning in Transformers (Allen-Zhu & Li, 2024), what formal languages they can learn (Strobl et al., 2024), their representational strengths and limitations (Sanford et al., 2023), how they learn shortcuts to automata (Liu et al., 2023), and how chain-of-thought provably improves their computation (Malach, 2023; Li et al., 2024b). However, this work remains disjoint from interpretability research, and approaches to mechanistic understanding often lack formal theory. This makes connecting high-level explanations to low-level processes and generalizing insights across architectures difficult. Meanwhile, multi-agent systems show empirical benefits but remain theoretically under-explored, particularly regarding how agents interact, which algorithms they perform, and how their strategies differ from end-to-end models. Overall, theoretical work on the algorithmic capacities of LLMs is still sparse. Future studies should investigate properties of good solutions (axiomatic desiderata), optimality and complexity proofs, and learning theory for LLM algorithms. A key question is whether certain architectures, parameters, and optimization schemes can guarantee the implementation of specific algorithms.

Algorithm evaluation should consider a number of relevant desiderata: *Fidelity*, which ensures consistent input–output reliability, *optimality*, which pursues the most efficient solutions (Shneidman & Parkes, 2004), and *minimality*, which values lower algorithmic complexity (Elmoznino et al., 2024). Further, algorithms should be *expressive* enough to be clearly understood and *runtime efficient* for scalability (Swartout & Moore, 1993). These properties form a basis for evaluating and developing algorithms in ML systems. An important direction is algorithm-centric architecture design and the reuse of verifiable algorithmic components, e.g., exploring symbolic operations to evoke specific algorithms. Another direction is to incentivize algorithmic building blocks during training, e.g., through complexity or sparsity regularization.

**Steering, guiding, optimization.** Optimizing a given LLM to learn or use specific algorithms can be steered via training, architecture, or ICL. Future work should examine how ICL can steer models toward a target algorithm and how feedforward or attention-based modifications improve primitives or composition. Another possibility is inference-time optimization to encourage specific algorithms. Ultimately, we can train or fine-tune LLMs to implement specific algorithms, guided by an algorithmic perspective.

**Designing new architectures.** A key future direction is algorithm-centric architecture design, which may be accomplished through the identification and re-use of verifiable algorithmic components. It would be especially interesting to see, for instance, if specific algorithms can be evoked through the use of symbolic operations, in context learning, or training regimes, e.g., GRPO (Shao et al., 2024) as opposed to Proximal Policy Optimization Algorithms or PPO (Schulman et al., 2017). An algorithmic understanding of architectural choices can improve how we build end-to-end architectures and multi-agent AI systems (Webb et al., 2024), paving the way for more transparent, reliable, and theoretically grounded generative AI. This enhanced algorithmic understanding can have significant implications for scientific applications of ML, potentially uncovering efficient strategies and fundamental mechanisms across various domain sciences.

**Conclusion.** While interpretability research has begun moving toward mechanistic and circuit-level analysis, it largely overlooks algorithmic explanations and evaluation of LLMs. **Our position is that the next generation of ML researchers should prioritize algorithmic understanding of generative AI.** We have presented AlgEval, approaches for algorithmic research, and actionable next steps. Deepening our understanding of how LLMs compute can enhance their sample efficiency, reduce emissions, and improve safety compliance, ultimately strengthening the overall impact of our community's work.

## Acknowledgements

The authors acknowledge Eder Sousa, Aishni Parab, Dalal Alharthi, Montassir Abbas, Andrea Kang, Hongjing Lu, Mark Green, Peter Todd, and the participants of the Institute for Pure and Applied Mathematics (IPAM) Long Program on the Mathematics of Intelligences. Part of this research was performed while some of the authors were visiting IPAM, which is supported by the National Science Foundation (Grant No. DMS-1925919).

## Impact Statement

We propose prioritizing a systematic understanding of algorithmic primitives and compositions in LLMs, and how they are learned and used given their architecture parameters, and training data. Below are a number of near-term and long-term impacts that can further motivate this research priority. We believe this research will impact more than mere understanding, and could potentially lead toward more efficient and lower-emission training and improving generative AI, and designing new architectures with algorithms in mind.

The rapid success and rise of generative AI faces the challenges of unpredictable errors as well as large emissions. In spite of these known challenges, the ML community's research priorities in the area of generative AI and LLMs have so far been myopically focused on improving performance, regardless of costs, and mechanistic interpretability, regardless of guarantees or algorithmic understanding. This position paper argues that the ML community should shift its focus to the algorithmic level of analysis, as the current priorities are often wasteful and offer only limited insight into models' fundamental inner workings.

A previous position paper on inner interpretability (Vilas et al., 2024), inspired by the analogy to Marr's levels in neuroscience (Marr, 1982), calls for attention to all levels: the computational level, the algorithmic level, and the implementation level. While we agree, we think there is a specific need to prioritize an algorithmic understanding in line with long-standing traditions in computer science. We highlight the importance of putting resources into understanding the algorithmic vocabulary and grammar of generative AI, which will contribute to the field beyond understanding isolated phenomena and toward a more systematic foundation.

### Sample Efficiency and Improved Behavior

An algorithmic understanding of generative AI can empower us to identify methods for improving sample-efficiency. This could occur through improving training with algorithmic performance in mind, optimizing and scaling inference-time compute and chain of thoughts with an understanding of their algorithmic implications. The latter could in turn improve the nebulous state of prompt engineering. Moreover, algorithm-based training and the reuse of algorithmic components in future models, both end-to-end and multi-agent, could lead to better behavioral performance and improved generalization. Ideally, this would yield fewer iterations for a given task, especially in multi-agent architectures. Together, sample efficiency during training and compute efficiency during response generation could potentially address both compute and emission challenges of generative AI.

### Emission Efficiency

Many assume generative AI's emission costs occur only during training, which equate to the lifetime emissions of multiple cars (Strubell et al., 2019). However, significant costs arise during use as well. For instance, a single interaction with a state-of-the-art LLM can require half a liter of water for cooling and emit the carbon equivalent of five gallons of gas when solving challenging problems with OpenAI's GPT-3. It is estimated that by 2027, global AI's water usage could rise to nearly two-thirds of the United Kingdom's annual consumption (Li et al., 2025). Repeated errors and back-and-forth interactions further increase these water and carbon costs, potentially leading to catastrophic climate impacts when scaled globally. Adopting algorithm-driven approaches can enhance emission efficiency in training and reasoning, and inform the design of new architectures. We believe that an algorithmic understanding of LLMs can lead to more environmentally sustainable methods, which is crucial as these models are rapidly integrated into everyday products.

We believe an algorithmic understanding of LLMs can lead to less wasteful approaches, considering the environmental costs of training and using generative AI. This is especially important given the rapid pace at which these models are being integrated into everyday products.

### Theories for Multi-Agent AI

Given the unreliability of most generative AI approaches, multi-agent LLM architectures have become common to correct LLM errors and hallucinations, orchestrate actions, and improve reasoning, among other use cases. While these approaches make LLM-based solutions more reliable, they lead to even higher emissions. On the other hand, various approaches are a patch-work of popular knowledge of psychology, and at best cognitive science or brain-inspired approaches, without any systematic theories or guarantees. One of the key challenges lies in building and analyzing agentic systems with provable performance, ensuring that they can function reliably and effectively in complex environments. We believe a better understanding of what LLMs truly do can lead to more efficient design of multi-agent systems as well as their implementation and product integration,

with advantages for engineers, users, and the planet.

### Trust, Compliance, and Safety

An enhanced understanding of model behavior enables researchers to discover novel mechanisms and advance a principled understanding of generative AI. An algorithmic understanding of LLMs can, in turn, increase the interoperability and trustworthiness of models. These insights can empower researchers and engineers to ensure compliance with safety standards.

### Algorithmic Bias

The challenge of bias in computer systems (Friedman & Nissenbaum, 1996), particularly instances of *technical bias* require an in-depth understanding of the underlying systems. In the context of generative AI, today's systems are commonly found to make unfair, undesired and even harmful predictions (Shah et al., 2020; Lucy & Bamman, 2021; Eberle et al., 2023; Fang et al., 2024), raising concerns about their deployment in sensitive domains. AlgEval directly supports the detection and understanding of algorithmic bias by systematically evaluating and interpreting a model's fundamental components. As bias can manifest at different levels within a system, for example, in low-level primitives, compositions thereof, or in the functioning of full algorithms, AlgEval offers a complementary approach to addressing pre-existing bias originating from training data. By targeting technical bias at these distinct levels, it enables a more granular and thorough evaluation of bias in generative models.

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

# A. Appendix

### A.1. Statistical Testing: Paired-sample *t*-tests for attention from the final token to correct vs. incorrect pathways.

Table 1: Reports statistics for the linear mixed-effects model, testing attention directed from the final token to the correct versus incorrect pathway across all layers. Mixed-effects modeling was conducted using the lmerTest package in R.

Table 2: Reports specific layers with significantly greater attention to the correct pathway, while Tab. 3 reports layers with significantly greater attention to the incorrect pathway. Note that conducting multiple t-tests (one for each layer) increases the risk of Type I errors (false positives). With $k$ layers, we have $k$ chances to incorrectly reject the null hypothesis. To mitigate the possibility of Type I errors, we used Bonferroni to control the family-wise error rate.

*Table 1.* Greater attention to the correct pathway across layers.

| *b* | *SE* | *t*-statistic | *df* | *p*-value |
|---|---|---|---|---|
| 0.33 | 0.07 | 4.51 | 2015 | $p = 7.0 \times 10^{-6}$ |

*Table 2.* Greater attention to the correct pathway (individual layers).

| Layer | *t*-Statistic | *df* | *p*-Value |
|---|---|---|---|
| 6 | 5.68 | 31 | $p = 1.5 \times 10^{-6}$ |
| 7 | 12.18 | 31 | $p = 1.17 \times 10^{-13}$ |
| 10 | 9.47 | 31 | $p = 5.86 \times 10^{-11}$ |
| 11 | 8.97 | 31 | $p = 1.99 \times 10^{-10}$ |
| 12 | 4.32 | 31 | $p = 7.49 \times 10^{-5}$ |
| 13 | 7.13 | 31 | $p = 2.60 \times 10^{-8}$ |
| 14 | 3.84 | 31 | $p = 2.87 \times 10^{-4}$ |
| 16 | 4.36 | 31 | $p = 6.65 \times 10^{-5}$ |
| 17 | 4.60 | 31 | $p = 3.40 \times 10^{-5}$ |
| 20 | 3.71 | 31 | $p = 4.1 \times 10^{-4}$ |
| 21 | 4.73 | 31 | $p = 2.36 \times 10^{-5}$ |
| 26 | 3.38 | 31 | $p = 9.9 \times 10^{-4}$ |
| 29 | 5.96 | 31 | $p = 6.97 \times 10^{-7}$ |
| 30 | 4.00 | 31 | $p = 1.83 \times 10^{-4}$ |

*Table 3.* Greater attention to the incorrect pathway (individual layers).

| Layer | *t*-statistic | *df* | *p*-value |
|---|---|---|---|
| 0 | -5.34 | 31 | $p = 4.05 \times 10^{-6}$ |
| 2 | -4.58 | 31 | $p = 3.52 \times 10^{-5}$ |
| 23 | -3.23 | 31 | $p = 1.47 \times 10^{-3}$ |

## A.2. Attention from the Goal Location to all Nodes for the Tree Graph (n=7)

**Prompts**

(1) From the hotel lobby, there are various rooms denoted by letter names. The lobby connects to O, which connects to W and Q. The lobby also connects to G, which connects to V and M. How can someone get to **W/Q/V/M** from the lobby?

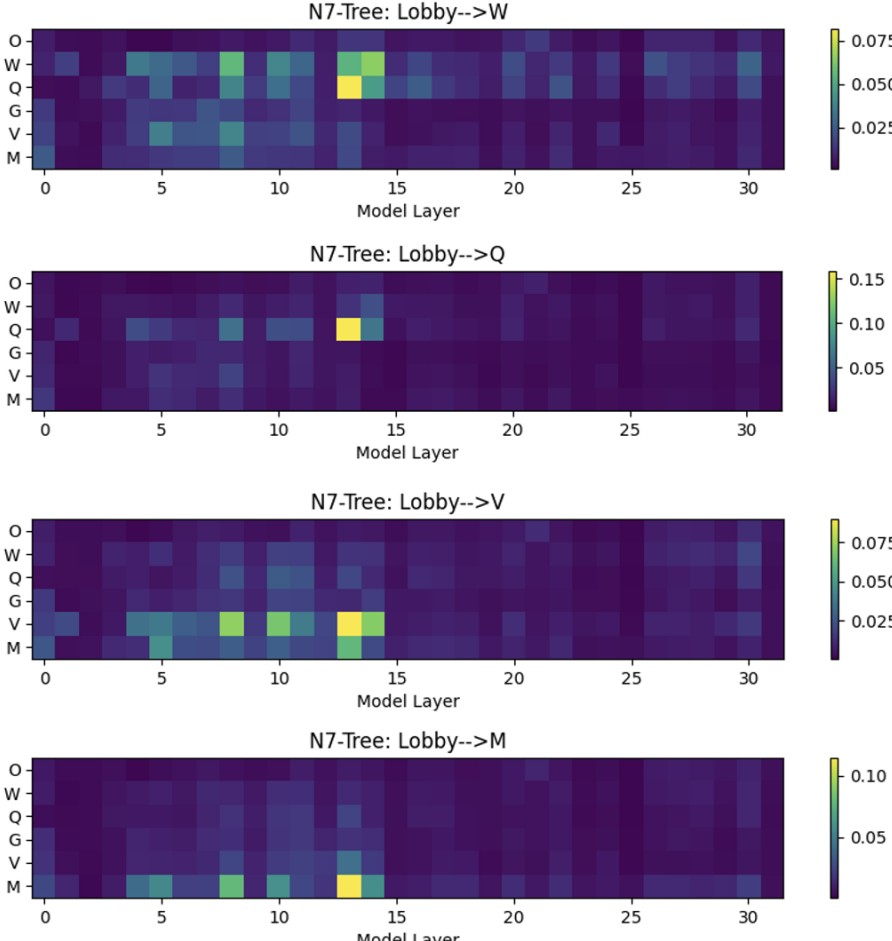

*Figure 4.* Attention heatmaps from the goal token to all graph nodes in the tree graph, when each final node is specified as the goal location.

### A.3. Analyzing Representations

**Prompts**

(1) Given a description of a set of rooms, determine if someone can get to W from the lobby? Only answer "Yes" or "No". The description is: From the hotel lobby, there are various rooms denoted by letter names. The lobby connects to O, which connects to W and Q. The lobby also connects to G, which connects to V and M. Answer:

(2) Given a description of a set of rooms, determine if someone can get to Q from the lobby? Only answer "Yes" or "No". The description is: From the hotel lobby, there are various rooms denoted by letter names. The lobby connects to O, which connects to W and Q. The lobby also connects to G, which connects to V and M. Answer:

(3) Given a description of a set of rooms, determine if someone can get to V from the lobby? Only answer "Yes" or "No". The description is: From the hotel lobby, there are various rooms denoted by letter names. The lobby connects to O, which connects to W and Q. The lobby also connects to G, which connects to V and M. Answer:

(4) Given a description of a set of rooms, determine if someone can get to M from the lobby? Only answer "Yes" or "No". The description is: From the hotel lobby, there are various rooms denoted by letter names. The lobby connects to O, which connects to W and Q. The lobby also connects to G, which connects to V and M. Answer:

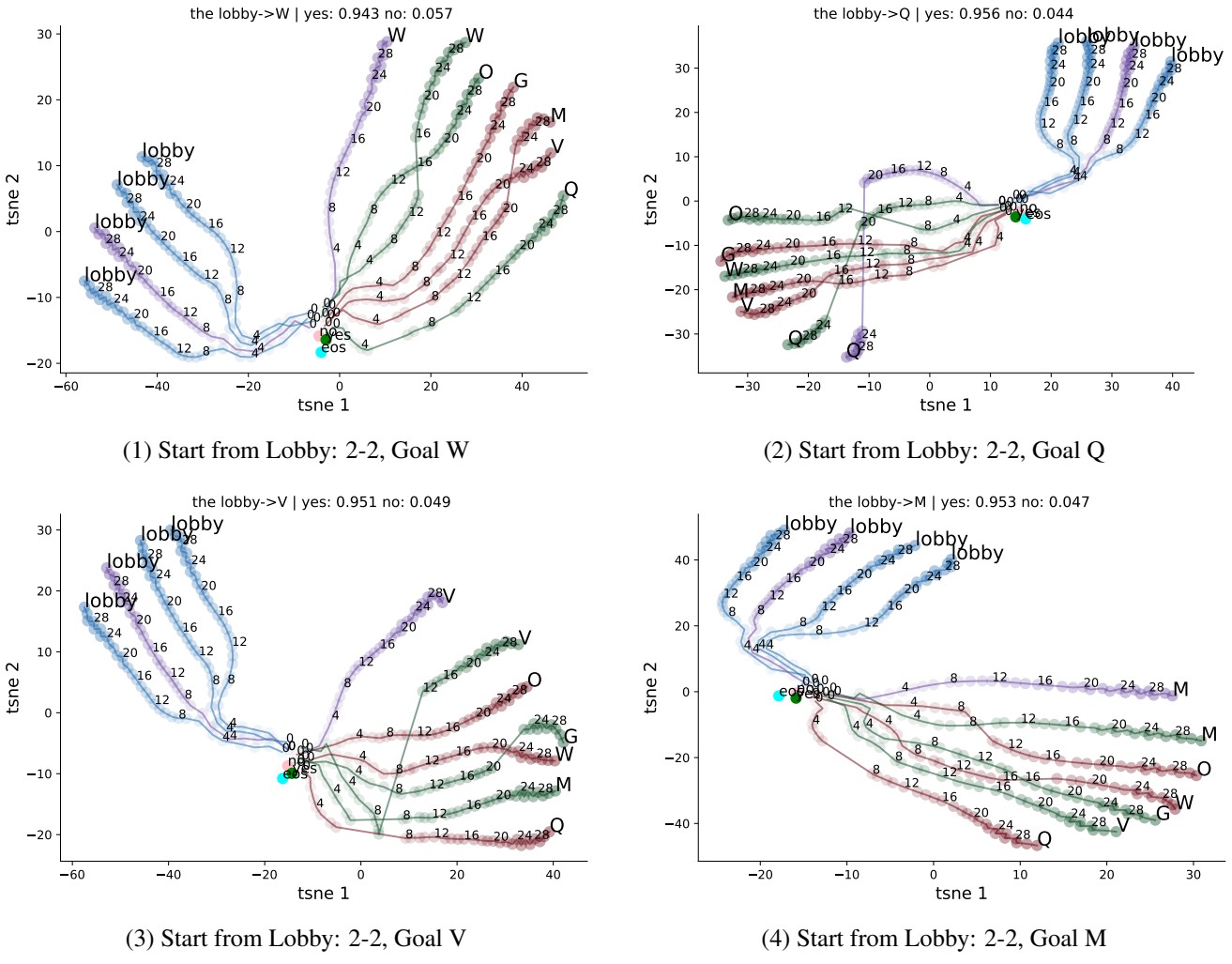

*Figure 5.* T-SNE projections of activations in a graph search setting. Plots correspond to prompts (1)–(4), corresponding to a changing goal node in a tree graph of n=7 nodes (see Figure 2). The probabilities of generating the correct 'yes' token as compared to the 'no' token using the final token representation is given for each setting.

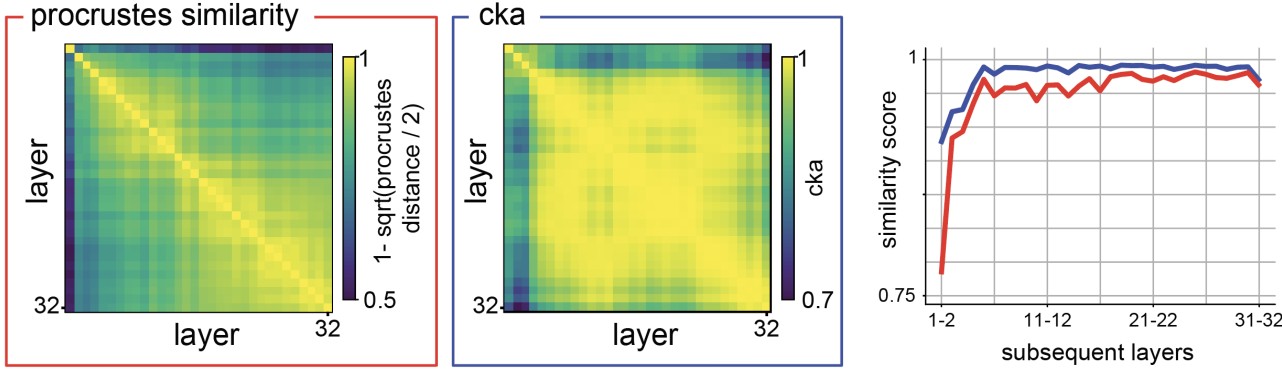

*Figure 6. Left:* We reasoned that discrete steps in a BFS or DFS rollout might be identified as substantial changes in representational geometry between subsequent layers. To explore this possibility, we computed representational similarity matrices using two similarity measures: the procrustes similarity (Williams et al., 2021) and the centered kernel alignment (Kornblith et al., 2019). *Right:* similarity scores across subsequent layers.

## A.4. Results on Llama3.1-70B-Instruct

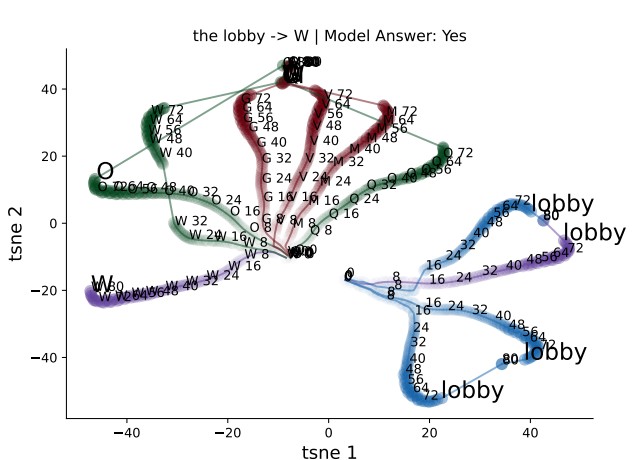

*Figure 7.* Representational analysis using Llama3.1-70B. Our t-SNE results show a clear separation of room representations across layers, in particular, a separation of non-goal nodes (green & red) vs. the goal node ('W' in purple). See also Figure 3 in the main paper for details.

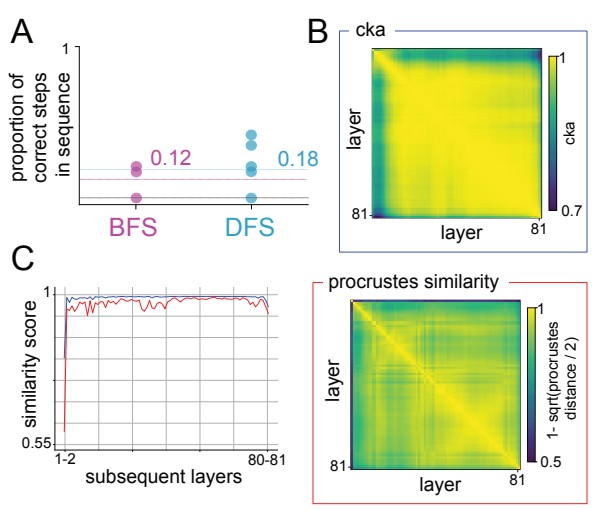

*Figure 8.* Evaluation of breadth-first search (BFS) and depth-first search (DFS) hypotheses in Llama3.1-70B. (**A**) Proportion of correct steps identified in the layer by layer hidden activations. Each data point represents a single rollout of the BFS or DFS algorithm. (**B**) Representational similarities between layers computed using two similarity measures. (**C**) Representational similarity between subsequent layers (off diagonal).

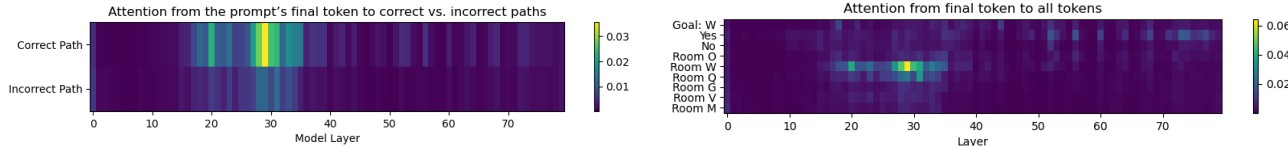

*Figure 9.* Attention heatmaps for Llama-3.1-70B. *Left:* Average attention per layer directed from the final token to the correct vs. incorrect pathways when node W is specified as the goal location. *Right:* Average attention from final token to different graph node and response tokens.

