# OpenReview forum: "Position: We Need An Algorithmic Understanding of Generative AI"
_ICML.cc/2025/Position_Paper_Track — ICML 2025 Position Paper Track spotlightposter_

### Official Review · Reviewer_Badu · 2025-02-21

**Significance:** 4
**Argument Clarity:** 4
**Rating:** 4
**Confidence:** 4

**Questions:**

NA

**Discussion Potential:**

4

**Paper Summary:**

This paper argues that we need to understand the algorithms (in a human-understandable way) that are implemented by trained neural networks. It argues that we need to understand single algorithm and how they compete against each other to reach the final answers.

## update after rebuttal
We thank the reviewer for their constructive and positive feedback. We’re honored they felt our Position articulates what the mechanistic interpretability field has long aimed to express.

**Position:**

Yes

**Position In Title:**

No

**Related Work:**

4

**Strengths And Weaknesses:**

The paper's strength is that it says what the field of mechanistic interpretability has always wanted to say and expressed it clearly. It has made solid arguments and presented case studies that the field should move forward towards discovering interpretable algorithms, AKA circuits, in these models.

**Support:**

4

---

> ### Author Rebuttal · Authors · 2025-03-31
>
> We would like to sincerely thank the reviewer for their constructive engagement with and positive evaluation of our work. It is an honor to read that our articulation of the position fits, in the reviewers words, with "what the field of mechanistic interpretability has always wanted to say and expressed it clearly".

---

### Official Review · Reviewer_JxdE · 2025-03-10

**Significance:** 3
**Argument Clarity:** 3
**Rating:** 4
**Confidence:** 3

**Questions:**

The paper proposes a top-down hypothesis-driven approach. Do the authors envision automated or semi-automated methods that can streamline the search for “algorithmic primitives” in large models?

**Discussion Potential:**

3

**Paper Summary:**

This paper argues that the community should prioritize an algorithmic understanding of generative AI models—particularly large language models (LLMs)—rather than relying primarily on scaling or on mechanistic interpretability alone. It proposes a framework called “AlgEval” for systematically studying the algorithms learned and used by LLMs. The paper reviews related work on mechanistic interpretability, attention, and representation analysis, and advocates for analyzing “algorithmic primitives” and how they compose into higher-level solutions. A brief case study on a simple graph-navigation task illustrates how one might investigate whether an LLM is using standard algorithms (e.g., BFS or DFS). The authors conclude by discussing open directions for theoretical work, bridging interpretability with algorithmic-level analyses, and potential benefits such as better sample efficiency, reduced environmental impact, and improved safety/compliance.

**Position:**

Yes

**Position In Title:**

Yes

**Related Work:**

3

**Strengths And Weaknesses:**

I am not an expert in this field, but this paper reminds me of the Physics of Language Models tutorial (https://physics.allen-zhu.com/). My comments on its strengths are as follows, from the perspective of a general machine learning researcher.

The paper explicitly advocates for an algorithmic investigation of LLMs—a timely and relevant stance for the ML community, especially given the growing focus on interpretability, safety, and efficiency. It effectively bridges mechanistic interpretability research (e.g., circuit analysis, attention, representational similarity) with classical algorithmic concepts such as BFS and DFS, offering a valuable synthesis of these often-disconnected research areas. The graph navigation example provides a concrete starting point for dissecting LLM behavior through hypothesized algorithms. While the analysis is relatively small in scope, it establishes a tangible framework for future empirical studies.

As for weaknesses, the paper could benefit from more intuitive illustrations to make its ideas more accessible to a broader audience. Given the breadth of related work and background on the algorithmic study of LLMs, readers unfamiliar with these topics may find the highly abstract summaries difficult to follow.

**Support:**

3

---

> ### Author Rebuttal · Authors · 2025-03-31
>
> We thank the reviewer for a constructive engagement with our paper. We have tried to address their comments.
> Please also see responses to reviewer rgcW. Our new Results can be downloaded here: https://tinyurl.com/AlgEval176
>
>
> - Intuitive illustration
> We weren't quite sure if the reviewer is asking for a figure, or further verbal explanations and examples. We are happy to provide either. In case the question is about more intuitive explanations of concrete steps, and in tandem with the reviewer's question about automation, we have composed the following response. Please let us know if it addresses your concerns.
>
> - AlgEval Steps and automation
>
> First, methodologically, we propose a 5-step process. 1- Identify and form a library of potential primitives based on known algorithms, which can grow over time and anchor corresponding tasks and mechanisms (described in the next 4 steps). Note that each algorithmic primitive can correspond to multiple tasks and algorithms, and can have different mechanistic implementations. Primitives can be both hypothesis-based, rooted in decades of theoretical research on algorithms, or empirically observed.
>
> 2- Identify a library of simple tasks, the solving of which requires a set of primitives. Examples of simple tasks include sequence induction [Olsson22], counting the integers between a specified lower and upper bound, outputting the mode for a sequence, or copying a sequence of unique tokens [Zhou24].
>
> 3- Create a library of mechanisms that implement primitives and corresponding analysis tools to identity them (which can be automated), e.g.,  layer by layer analysis as in the case study, function vectors [Todd24], binding in attention heads [Feng24], or layer-wise attributions and assumptions on the data-generating function [Wied23]. These tools can be inspired by mech interpretation, neuroscience, information theory, complexity theory, behavioral sciences, physics/chemistry (e.g., similar methods for atoms and molecules), etc.
>
> 4- Composition: An important aspect of identifying primitives involves how they are composed together, how the larger algorithm is implemented across model layers and inference time compute, how recursion might be implemented using the same primitive, and whether and to what extent the compositional tendencies (i.e., which primitives are composed together and how) generalize across tasks and models (e.g., DeepSeek might have a different strategy due to the use of RL in its training, and a Tower of Hanoi might use a different search algorithm compared to planning a vacation).
>
> 5- Ablations: A set of tools for causal intervention and ablation [Talon24] to identify the primitive’s role. Newly discovered primitives, and their corresponding tasks, mechanisms, and methods, can then be added to the primitives library.
>
> For automated discovery:
>
> - It is possible to generate multiple hypotheses and test each hypothesis iteratively across all attention heads, layers, or activations of tokens across layers—among other methods.
>
> - Decomposing the evaluation of hypotheses across different mechanistic components of the model, such as attention heads or representations, can help automate the evaluation of their individual and combined roles.
>
> - While in the early phases of AlgEval it is important to identify tasks with careful methods and expert supervision (e.g., inspired by behavioral and cognitive sciences), it is in principle possible to automate task identification as well, e.g., simply using existing tasks in evaluation benchmarks, e.g. similar to clustering model strategies based on input heatmaps in image classification [Lap19].
>
> - Creating a compositional task library as well as similarity measures across tasks, will allow us to automate "a search across task space" to more systematic identify relevant potential tasks tailored to specific algorithms or primitives.
>
> We thank the reviewer again. These is a preliminary sketch of concrete steps for the broader AlgEval framework.
>
>
>
> --------  References  --------
>
> [Olsson22] Olsson, C., et al. (2022). In-context learning and induction heads. Transformer Circuits Thread.
>
> [Zhou24] Zhou, H., et al. What algorithms can transformers learn? A study in length generalization. ICLR 2024.
>
> [Todd24] Todd, E., et al. (2023). Function vectors in large language models. ICLR 2024.
>
> [Feng24] Feng, J., & Steinhardt, J. How do Language Models Bind Entities in Context? ICLR 2024.
>
> [Wied23] T. Wiedemer,  et al. Compositional generalization from first principles. NeurIPS 2023.
>
> [Talon24] Talon, D. et al. Towards the reusability and compositionality of causal representations. PMLR vol 236:1–22, 2024 Conf. on Causal Learning and Reasoning, 2024.
>
> [Lap19] Lapuschkin, S., et al. "Unmasking Clever Hans predictors and assessing what machines really learn." Nature communications 10.1 (2019): 1096.

---

### Official Review · Reviewer_rgcW · 2025-03-13

**Significance:** 3
**Argument Clarity:** 3
**Rating:** 3
**Confidence:** 3

**Questions:**

Your paper advocates for an algorithmic understanding of LLMs, which strongly relates to Neural Algorithmic Reasoning (NAR).
Many NAR studies involve training neural networks to explicitly mimic known algorithms. Do you see potential in adapting such approaches to reverse-engineer the emergent algorithms inside LLMs?

How should we formally define and detect primitives? In traditional program synthesis, models compose modular functions to generate structured outputs.

**Discussion Potential:**

4

**Paper Summary:**

The paper criticizes the current state of research on LLMs, which prioritizes scaling over understanding. It argues that:

- LLMs exhibit emergent algorithmic behavior, but the field lacks a systematic way to study it.
- Existing interpretability research focuses on individual model components rather than algorithmic structures.
- Blindly scaling LLMs without understanding their underlying computations is inefficient and leads to high resource consumption.

To address this, the paper introduces AlgEval, a framework for:
- Identifying algorithmic primitives (e.g., memory retrieval, sorting, arithmetic).
- Understanding how LLMs compose these primitives into algorithms (e.g., search, planning).
- Studying inference-time computation, including chain-of-thought reasoning and multi-agent interactions.

### update after rebuttal
Thanks for the clarification. These discussions might help future extensions regarding the specific form of derive algorithms.
Also, the tinyurl link does not work for me.
In my opinion, I believe the current score is sufficient (I believe further development is needed to address my questions).

**Position:**

Yes

**Position In Title:**

Yes

**Related Work:**

3

**Strengths And Weaknesses:**

**Strengths**
- The paper highlights a major issue that LLMs may be learning algorithms, but researchers don’t fully understand them.
It proposes a framework (AlgEval) for addressing this issue in a structured way.
- The paper clearly differentiates algorithmic understanding from interpretability. It provides a well-structured breakdown of algorithmic primitives, composition, and inference-time compute.
- The graph search task demonstrates that LLMs do not strictly follow classical algorithms, reinforcing the need for algorithmic evaluation.


**Weaknesses**

- The case study is limited in scope (only one LLM tested on one task).
- No clear methodology for defining algorithmic primitives: While the paper proposes identifying algorithmic primitives, it does not provide a systematic method for discovering them.
- The paper calls for prioritizing algorithmic understanding, but does not specify what concrete steps the research community should take.

**Support:**

3

---

> ### Author Rebuttal · Authors · 2025-03-31
>
> We first thank the reviewer for their constructive feedback, allowing us to making ALgEval steps more concrete.
>
> Weakness 1. We fully agree. Please see our new results, in response to reviewer tuDy.
> New Result figures can be downloaded here: https://tinyurl.com/AlgEval176
>
> Weakness 2, 3, and Question 2.
> We sincerely thank the reviewer for this question. It allowed us to further expand on what we foresee in AlgEval.
>
> Algorithmic primitives are building blocks of algorithms, which canNot be reduced to more basic interpretable algorithms, but which can be implemented using different mechanisms. An example is “negation”, it is a basic part of many algorithms, but mechanistically it can be implemented via attention heads, function vectors, a basis, etc., a repertoire of mechanisms can all implement the same algorithmic primitive.
> Methodologically and in terms of concrete steps, we propose 5 steps.
>
> 1- Identify and form a library of potential primitives based on known algorithms, which can grow over time and anchor corresponding tasks and mechanisms (described in the next 4 steps). Note that each algorithmic primitive can correspond to multiple tasks and algorithms, and can have different mechanistic implementations. Primitives can be both hypothesis-based, rooted in decades of theoretical research on algorithms, or empirically observed.
>
> 2- Identify a library of simple tasks, the solving of which requires a set of primitives. Examples of simple tasks include sequence induction [Olsson22], counting the integers between a specified lower and upper bound, outputting the mode for a sequence, or copying a sequence of unique tokens [Zhou24].
>
> 3- Create a library of mechanisms that implement primitives and corresponding analysis tools to identity them (which can be automated), e.g.,  layer by layer analysis as in the case study, function vectors [Todd24], binding in attention heads [Feng24], or layer-wise attributions and assumptions on the data-generating function [Wied23]. These tools can be inspired by mech interpretation, neuroscience, information theory, complexity theory, behavioral sciences, physics/chemistry (e.g., similar methods for atoms and molecules), etc.
>
> 4- Composition: An important aspect of identifying primitives involves how they are composed together, how the larger algorithm is implemented across model layers and inference time compute, how recursion might be implemented using the same primitive, and whether and to what extent the compositional tendencies (i.e., which primitives are composed together and how) generalize across tasks and models (e.g., DeepSeek might have a different strategy due to the use of RL in its training, and a Tower of Hanoi might use a different search algorithm compared to planning a vacation).
>
> 5- Ablation: A set of tools for causal intervention and ablation [Talon24] to identify the primitive’s role. Newly discovered primitives, and their corresponding tasks, mechanisms, and methods, can then be added to the primitives library.
>
> Question 3.
>
> NAR focused on “building neural networks that are able to execute algorithmic computation” (Neuron 2021). AlgEval is primarily focused on Evaluation and identification of actual algorithms that a model ends up using in practice, while instilling algorithms or building neural networks that can use them are secondary goals. Thus, we think in the future AlgEval can benefit from approaches like NAR as one of the tools for ensuring efficient implementation of desired algorithms by models—either by architecture design, or training regime.
>
>
> -------- References -------------
>
> [Olsson22] Olsson, C., et al. (2022). In-context learning and induction heads. Transformer Circuits Thread.
>
> [Zhou24] Zhou, H., et al. What algorithms can transformers learn? A study in length generalization. ICLR 2024.
>
> [Todd24] Todd, E., et al. (2023). Function vectors in large language models. ICLR 2024.
>
> [Feng24] Feng, J., & Steinhardt, J. How do Language Models Bind Entities in Context? ICLR 2024.
>
> [Wied23] T. Wiedemer,  et al. Compositional generalization from first principles. NeurIPS 2023.
>
> [Talon24] Talon, D. et al. Towards the reusability and compositionality of causal representations. PMLR vol 236:1–22, 2024 Conf. on Causal Learning and Reasoning, 2024.

---

### Official Review · Reviewer_tuDy · 2025-03-15

**Significance:** 4
**Argument Clarity:** 4
**Rating:** 5
**Confidence:** 4

**Questions:**

1. Could you elaborate on specific tasks or domains where algorithmic understanding might provide the most immediate or substantial benefits?
2. What are the limitations or potential challenges in scaling the AlgEval framework to state-of-the-art LLMs (e.g., GPT-4, Gemini)?
3. Can you provide a clearer comparison between AlgEval and existing interpretability frameworks, highlighting the distinct advantages of AlgEval in practice?

**Discussion Potential:**

4

**Paper Summary:**

This paper argues that the machine learning community should prioritize algorithmic understanding of generative AI, particularly large language models (LLMs). It introduces AlgEval, a systematic framework for uncovering algorithmic primitives (basic computational components) and algorithmic compositions (how these primitives combine to solve tasks) within LLMs. Through detailed attention and representational analyses, the paper illustrates how internal mechanisms of LLMs could be systematically evaluated, providing insights into the actual algorithms that models implicitly learn. The paper further discusses how this approach can lead to more efficient training methods, improved model interpretability, sustainability, and new architecture designs.

**Position:**

Yes

**Position In Title:**

Yes

**Related Work:**

4

**Strengths And Weaknesses:**

Strengths:
- Clearly stated and highly relevant position that addresses a significant research gap.
- Well-structured methodological proposal (AlgEval) with clearly defined algorithmic primitives and compositions.
- Comprehensive review and thoughtful integration of related literature in interpretability and mechanistic understanding.
- Presents a concrete empirical case study demonstrating the feasibility and value of the proposed framework.
- Strong arguments regarding potential impacts on efficiency, interpretability, and environmental sustainability.

Weaknesses:
- The empirical case study, while illustrative, is limited in scope and may not fully validate the practicality of AlgEval across broader, more complex scenarios.
- Discussion of alternative views and theoretical considerations could be expanded further to address potential critiques.
- Some theoretical claims regarding the potential impact on model efficiency and emission reductions could benefit from additional supporting evidence or concrete examples.

**Support:**

4

---

> ### Author Rebuttal · Authors · 2025-03-31
>
> We sincerely thank the reviewer for a meticulous and encouraging engagement with our work.
>
> Weakness 1.
>
> We agree with the reviewer that one LLM and task are limited, even for a position paper. Within the brief rebuttal period we expanded the case study to Llama 3.1 70B. Results can be downloaded here: https://tinyurl.com/AlgEval176
>
> In short: our representation analysis shows a similar pattern, with more pronounced separation of correct and incorrect trajectories in the latent activation state space (Fig 2 in the new PDF). Moreover, we compared the sequence of states with highest representation activation across the layers to possible search sequences generated by BFS and DFS strategies. We did not find any differences in search strategy between the models (neither sequence of hidden state representations is a good match for BFS/DFS rollouts on the target graph). In sum, a model with ten times more parameters was not more likely to use one of the classic search algorithms we compared. Both models seem to advance based on changes in representational similarity: increasing the distance to less relevant policies as if discarding them, however, their strategy doesn’t obviously match BFS to DFS. Future work is required to increase the number of tasks and models, test more algorithms, how representations and attention implement primitives (even unconventional ones), and investigate how search can be implemented in terms of gradual changes in the latent representation space.
>
> Weakness 2, 3.
>
> In the camera ready version we will gladly expand section 6 and the discussion of other interpretability techniques for interpretability methods, from game theoretic to program synthesis, as well as broader considerations from theoretical CS, such as complexity theory. We will also compare to NAR and physics of LLMs noted by other reviewers. Similarly, we will extend supporting evidence and concrete examples in our claims regarding the potential impact on model efficiency and emission reductions. If the reviewer has specific theoretical accounts in mind we would love to receive pointers and are happy to incorporate them.
>
> Question 1. Three category of tasks can be prioritized.
>
> 1- Simple tasks to identify primitives
> Examples include analogical reasoning, transitive inference A->B, B->C, function vector type things but rooted in ground truth, sanity checks—many of which leverage a geometry of sorts. Other simple tasks include: count, mode, copy with unique tokens, negation, and sort.
> (Harder, but still simple): copy with repeat tokens, addition, and parity.
> 2- Frontier tasks such as multi-step or complex reasoning
> Tasks includes generating factual replies, mathematical reasoning, code synthesis, generating compositional images, generating proteins with specific properties,...
> 3- High-impact tasks
> These tasks can be simple or complex, but they are high in demand, either by users, improve security, could improve welfare, etc.
>
> Question 2.
>
> Many of the SoTa models (including those the reviewer mentions) are proprietary. Serious challenges therefore include: No access to weights, activations, attentions. No access to inference time compute in full. Little control over the model, no access to architecture for ablations.
> But this might not be a hard limitation: e.g. open models are catching up and DeepSeek has comparable size to the larger models. This challenge is not a conceptual one, but an access issue.
> Another Potential limitation for scaling to larger models is high compute, which is especially a challenge for automated analysis, but that is also not a hard limitation. In fact, in the case of potentially intractable analysis with in principle some combinatorial explosion => algEval might give guidance on how to break the complexity down. This might require more "scientist in the loop" rather than fully automated discovery.
>
> Question 3.
> Great question! In addition to what we discussed in the paper, we address references suggested by other reviewers.
>
> 3.a. NAR: focused on “building neural networks that are able to execute algorithmic computation” (Neuron 2021). AlgEval is primarily focused on Evaluation and identification of actual algorithms that a model ends up using in practice, rather than instilling algorithms or building neural networks that can use them. The latter is a secondary goal of AlgEval, and AlgEval can benefit from approaches like NAR as one of the tools.
>
> 3.b. Mechanistic Interpretability: primary focus on circuits rather than primitives or algorithms or primitives and their composition, evaluation and theoretical perspectives are often an afterthought, however AlgEval uses their methods for identifying/evaluating primitives and composition.
>
> 3.c.Another comparison is to Distilling bayesian priors (Brenden Lake), which also differs from analyzing algorithms or primitives that models truly use. In the interest of space we stop here, but are more than happy to incorporate views the reviewer deems relevant.

---

### Decision · Program_Chairs · 2025-04-30

**Decision:**

Accept (spotlight poster)

**Comment:**

All reviewers recommend accept. They appreciate the clearly stated position, the timeliness and relevance of the argument, as well as the demonstrated case study. The AC feels that this is a strong submission and a good position paper.